# Pulmonary sequelae of SARS-CoV-2 infection and factors associated with persistent abnormal lung function at six months after infection: Prospective cohort study

Bashour Yazji[1], Nha Voduc[2], Sunita Mulpuru[1,2], Juthaporn Cowan[1,3,4]*

1 The Ottawa Hospital Research Institute, Ottawa, Ontario, Canada, 2 Division of Respirology, Department of Medicine, University of Ottawa, Ontario, Canada, 3 Division of Infectious Diseases, Department of Medicine, University of Ottawa, Ontario, Canada, 4 Centre of Infection, Immunity and Inflammation, University of Ottawa, Ontario, Canada

* jcowan@toh.ca

## Abstract

### Background

Information on the long-term pulmonary sequelae following SARS-CoV-2 infection is limited.

### Methods

Prospective cohort study of hospitalized and non-hospitalized adult patients age >18 with documented SARS-CoV-2 infection by RT-PCR three months prior to enrolment between June and December 2020. Participants underwent full pulmonary function test (PFT), cardiopulmonary exercise testing at 3 months and 6 months. Primary outcome was mean differences of forced vital capacity (FVC), diffuse capacity of lung for carbon monoxide (DLCO), and oxygen consumption (VO2) at 6 vs. 3 months. Secondary outcomes were respiratory outcomes classified into 5 clinical groups–no lung disease, resolved lung disease, persistent lung disease, PFT abnormalities attributable to pre-existing lung disease or other factors, and mild PFT abnormalities of uncertain clinical significance.

### Results

Fifty-one, 30 hospitalized and 21 non-hospitalized, participants were included. Median age was 51 years; 20 (39.2%) were female. Mean (±SD) percent predicted values of FVC, DLCO and VO2 at 3 vs 6-month-visits were 96.2 ± 15.6 vs. 97.6 ± 15.5, 73.74 ±18 vs. 78.5 ± 15.5, and 75.5 ± 18.9 vs. 76.1 ± 21.5, respectively. Nineteen (37%) patients had physiologic and/or radiographic evidence of lung disease at 3 months with eight (15.7%) continuing to have persistent disease at 6 months. History of diabetes, hypertension, ICU admission and elevated D-Dimer levels were associated with persistent lung disease at 6 months.

**Data Availability Statement:** All relevant data are within the paper and its Supporting Information files.

**Funding:** J.C: $25,000 CAD The Ottawa Hospital COVID-19 Emergency Response Fund. The funders had no role in study design, data collection and analysis, decision to publish, or preparation of the manuscript.

**Competing interests:** The authors have declared that no competing interests exist.

## Interpretation

Persistent lung disease at 6 months post SARS-CoV-2 infection exists. Changes of lung function between 3- and 6-months are not significant. A longer follow-up is required to determine long-term prognosis.

## Introduction

Respiratory tract is an initial site of severe acute respiratory syndrome coronavirus-2 (SARS-CoV-2) infection which lead to Coronavirus Disease 2019 (COVID-19) [1, 2]. Pneumonia is the predominant clinical presentation, among others [3, 4]. Persistent COVID-19 symptoms or post-COVID condition has been reported [5]. The most common persistent symptoms include fatigue, breathlessness, and altered cognitive function [5, 6]. However, there is limited information on the long-term pulmonary sequelae measured by pulmonary function test. Most studies evaluating post-COVID lung function assess patients 1–3 months post-infection [7–9]. The available long-term data is limited to hospitalized patients [10, 11]. We previously reported residual symptoms, mild impairment of lung volumes, gas exchange and a diminished exercise capacity at four months post SARS-CoV-2 infection regardless of severity during acute infection [12].

In this study, we aimed to describe the clinical and physiological respiratory trajectories among patients with COVID-19 at 6 months following the infection and sought to examine risk factors for persistent COVID pulmonary disease at 6 months post infection.

## Methods

### Study design

We conducted a prospective cohort study of hospitalized and non-hospitalized patients with COVID-19. The enrolment period was June to December 2020.

### Population

The patients were screened and recruited from The Ottawa Hospital administrative registry and from the study website (https://omc.ohri.ca/left/). Patients were included in the study if they were ≥18 years old and diagnosed with a COVID-19 infection by RT-PCR 3 months (+6 weeks) before enrollment. Patients with history of pulmonary resection, mobility issues that may interfere with cardiopulmonary exercise test, or pregnancy were excluded. The study was approved by the Ottawa Health Science Network Research Ethics Board (Protocol number 20200371-01H).

### Data collection

Patients underwent pulmonary function testing (PFT), and symptom-limited incremental (15 Watts /minute) cardiopulmonary exercise testing (CPET) at 3 months (+6 weeks) and 6 months (+6 weeks) following COVID-19. Pulmonary function testing (Vmax systems; Sensor-Medics) was performed in accordance with American Thoracic Society standards and included spirometry, lung volume and diffusing capacity measurements. PFT results were reported as a percentage predicted of established reference values. Cardiopulmonary exercise testing was performed on an electronically braked cycle ergometer with computerized CPET system (Vmax229d; Sensormedics). Jones reference equations were used to determine predicted oxygen consumption (VO2) maximums for each subject. At each study visit, patients

also completed St. George Respiratory Questionnaire (SGRQ) and Short Form Survey (SF-12). Demographics, co-morbidities and smoking history were recorded at time of enrolment. COVID-19 infection severity during the acute phase was graded from 1 to 6. Grade 1 was defined as no medical intervention; 2 was Emergency Department (ED) visit; 3 was hospitalization but no use of supplemental oxygen; 4 was hospitalization requiring low-flow supplemental oxygen $\leq$ 10 L/min; 5 was hospitalization requiring high-flow heated oxygen; and 6 was mechanical ventilation. Blood samples were collected for analysis of inflammatory markers including C-reactive protein (CRP), erythrocyte sedimentation rate (ESR), and D-dimer. The radiology reports (chest x-ray and CT chest imaging) obtained any time between initial COVID infection and 6-month follow-up were collected and abnormal findings were confirmed by NV.

**Primary outcome.** Mean differences of forced vital capacity (FVC), diffusing capacity of lung for carbon monoxide (DLCO), and VO2 at 6 vs. 3 months.

**Secondary outcomes.** Clinical and physiological trajectories of respiratory outcome measures. All patients with PFT abnormalities were referred to a respirologist based at the Ottawa Hospital for further evaluation. A standardized radiographic imaging protocol was not part of our study methodology as resource limitations during the early part of the COVID pandemic precluded routine radiographic imaging for patients with no physiologic abnormality. However, we conduct a thorough chart review of relevant chest radiographic data (chest x-ray and chest CT images) for all study patients and all patients with PFT abnormalities did undergo at least CXR imaging ordered by the consulting respirologist, if this was not already done.

We classified patients into the following 5 clinical groups based on a combination of reported clinical symptoms, PFT, chest radiographs, and ventilatory responses during cardiopulmonary exercise testing.

1. No lung disease. Patients with normal pulmonary function testing at 3 and 6 months and no abnormalities in ventilatory responses during cardiopulmonary exercise testing. Imaging was not always available but there were no radiographic abnormalities present on any available imaging in this group.

2. Resolved COVID lung disease. Patients with PFT abnormalities and radiographic evidence of disease either at the time of COVID-19 diagnosis or at the 3-month visit, but normal lung function at time of 6-month follow-up.

3. Persisting COVID lung disease. Patients with PFTs and/or radiographic abnormalities compatible with COVID-19 related parenchymal lung disease at 6 months.

4. Pulmonary function abnormalities not related to COVID-19. Patients had PFT abnormalities without radiographic evidence of COVID-19 related parenchymal lung disease at any time during the study period. Additionally, clinical assessment and chart review suggested that another condition was the most likely cause of the observed PFT abnormalities. These conditions included asthma, chronic obstructive pulmonary disease (COPD), pulmonary embolism, obesity and anemia.

5. Mild PFT abnormalities of uncertain clinical significance. Patients with PFT abnormalities without radiographic evidence of COVID-19 and no prior clinical diagnosis of lung or cardiac disease.

## Analysis

We used descriptive statistics (means, standard deviation; medians, interquartile range; number, proportions) to describe the characteristics of the study cohort and compare clinical

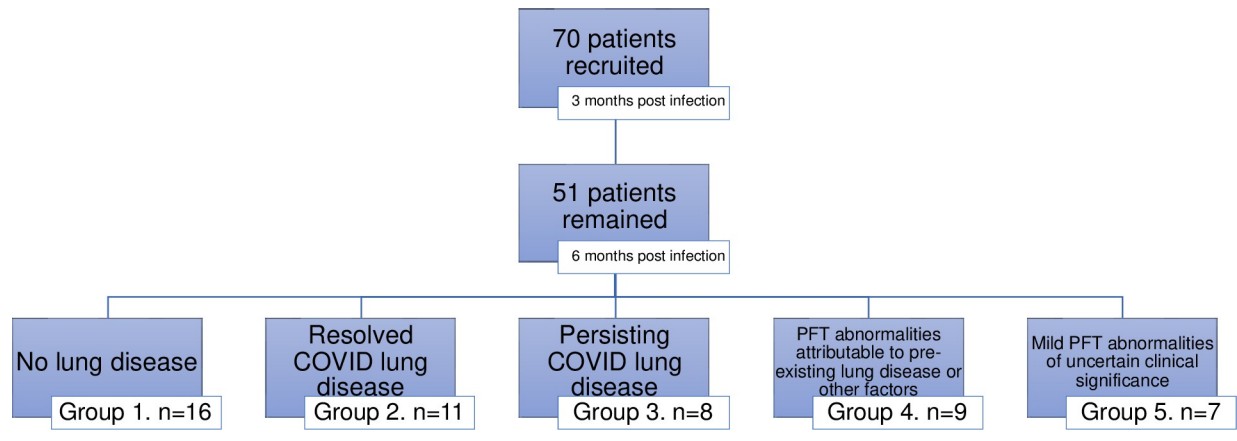

**Fig 1. Study participants flow chart.**

variables between groups. We used univariate logistic regression to explore the association between clinical variables and persisting COVID lung disease at 6 months post infection.

## Results

Of 70 patients recruited, 51 patients had complete data and thus were included in this analysis [Fig 1]. The mean time from baseline to the 3- and 6-month study visits were 15.5 ± 1.9 weeks and 28.7 ± 1.8 weeks, respectively.

Table 1 describes the demographics, comorbidities, health service use, and lung function parameters of the patient cohort. Median age was 51 years; 20 (39.2%) were female, 13 (25.5%) were obese, and 18 (35.3%) were active smokers. Thirty patients did not require hospitalization (30/51, 58.8%) and 16 (31.4%) had a self-reported history of lung disease.

Mean (± SD) percent predicted values of FVC, DLCO and VO2 at 3 vs 6-month-visit were 96.2 ± 15.6 vs. 97.6 ± 15.5, 73.74 ±18 vs. 78.5 ± 15.5, and 75.5 ± 18.9 vs. 76.1 ± 21.5 [Fig 2]. Changes in lung function measurements between 3- and 6-month visits were not statistically significant.

Based on a comprehensive evaluation using clinical chart review (including consultation by Respirologists), PFT, chest radiographs and CPET data, patients were classified into 5 clinical outcomes [Table 1 and Fig 1]. There were 16 (31.3%) with no lung disease, 11 (21.5%) with resolved COVID lung disease, 9 (17.6%) had PFT abnormalities attributable to pre-existing lung disease or other factors, and 7 (13.7%) had mild PFT abnormalities of uncertain clinical significance. 8 patients (15.7%) had persistent COVID lung disease.

Thirteen of 16 patients in group 1 (normal lung function) had no radiographic imaging during the study period. All patients in other groups had at least chest x-ray imaging. Eighteen had CT imaging during the study period, including all 8 patients in group 3 (persisting PFT abnormalities). Chest x-ray findings ranged from normal chest-ray to interstitial opacities to confluent airspace opacities. Initial CT findings ranged from normal to ground glass opacities to patchy consolidation, compatible with organizing pneumonia. Follow-up CT imaging of the 8 patients with persisting PFT abnormalities, demonstrated no parenchymal lung disease in 3 cases and findings ranging from mild ground glass opacities to early fibrosis with reticular opacities and traction bronchiolectasis in the other 5 cases.

All 8 patients with persisting COVID lung disease were hospitalized, 50% (4/8) were admitted to the ICU, and none were actively smoking at the time of study entry.

Table 2 summarizes the unadjusted association between clinical factors and persistent COVID lung disease at 6 months. Patients requiring ICU admission were more likely to have

**Table 1. Characteristics of the study cohort.**

| Variable | Total cohort N = 51 | Group 1: No lung disease N = 16 | Group 2: Resolved COVID lung disease N = 11 | Group 3: Persisting COVID lung disease N = 8 | Group 4: PFT abnormalities attributable to pre-existing lung disease or other factors N = 9 | Group 5: Mild PFT abnormalities of uncertain clinical significance N = 7 |
|---|---|---|---|---|---|---|
| Age > = 65 years (n,%) | 9 (17.6) | 1 (6.3) | 3 (27.3) | 2 (25) | 2 (22.2) | 1 (14.3) |
| Age (median, IQR) | 51 (21.5) | 46.5 (16) | 61 (16) | 61 (12) | 49 (38.5) | 47 (19) |
| Female Sex (n, %) | 20 (39.2) | 7 (43.7) | 2 (18.2) | 3 (37.5) | 3 (33.3) | 5 (71.4) |
| BMI kg/m$^2$ (median, IQR) | 28 (54.9) | 29.1 (5.5) | 28.2 (4.3) | 26.15 (9.45) | 26.5 (5.35) | 26.3 (3.6) |
| BMI kg/m$^2$ > = 30 (n, %) | 13 (25.5) | 6 (37.5) | 3 (27.3) | 2 (25) | 1 (11.1) | 1 (14.3) |
| Active Tobacco Smoking (n,%) | 18 (35.3) | 8 (50) | 5 (45.4) | 0 (0) | 2 (22.2) | 3 (42.9) |
| COVID Severity Scale Score: 1–6 (n,%)[a] | | | | | | |
| 1 | 19 (37.2) | 9 (56.2) | 1 (9) | 0 (0) | 5 (55.5) | 4 (50) |
| 2 | 11 (21.5) | 7 (43.7) | 1 (9) | 0 (0) | 1 (11.1) | 2 (28.6) |
| 3 | 11 (21.6) | 0 (0) | 5 (45.4) | 4 (50) | 2 (22.2) | 0 (0) |
| 4 | 0 (0) | 0 (0) | 0 (0) | 0 (0) | 0 (0) | 0 (0) |
| 5 | 5 (9.8) | 0 (0) | 2 (18.2) | 2 (25) | 0 (0) | 1 (14.3) |
| 6 | 5 (9.8) | 0 (0) | 2 (18.2) | 2 (25) | 1 (11.1) | 0 (0) |
| D-Dimer[b] > = 250 ng/mL (n,%) | 23 (46) | 3 (18.7) | 7 (63.6) | 7 (87.5) | 4 (44.4) | 2 (28.6) |
| Pre-Existing Lung Diseases (n, %) | 16 (31.4%) | 3 (18.7) | 5 (45.4) | 2 (25) | 4 (44.4) | 2 (28.6) |
| Heart Failure/CAD (n,%) | 2 (3.9) | 0 (0) | 0 (0) | 1 (12.5) | 0 (0) | 1 (14.3) |
| Hypertension (n,%) | 11 (21.6) | 1 (6.2) | 2 (18.2) | 4 (50) | 3 (33.3) | 1 (14.3) |
| Diabetes (n,%) | 8 (15.7) | 0 (0) | 2 (18.2) | 4 (50) | 1 (11.1) | 1 (14.3) |
| Chronic Kidney Disease (n,%) | 2 (3.9) | 0 (0) | 0 (0) | 1 (12.5) | 1 (11.1) | 0 (0) |
| Cancer (n,%) | 9 (17.6) | 0 (0) | 3 (27%) | 2 (25%) | 4 (44.4) | 0 (0) |
| Worsened HRQOL score from 3 to 6 months post COVID infection (SGRQ score)[d] (n,%) | 21 (42%) | 6 (37.5) | 5 (45.4) | 3 (37.5) | 6 (66.6) | 1 (14.3) |
| Worsened HRQOL by SGRQ score > = 4 points, from 3–6 months post COVID infection (n,%) | 10 (20%) | 2 (12.5) | 3 (27.3) | 1 (12.5) | 4 (44.4) | 0 (0) |
| Hospitalization (n,%) | 21 (37.2) | 0 (0) | 9 (81.8) | 8 (100) | 3 (33.3) | 1 (14.3) |
| ICU Admission (n, %) | 10 (19.6) | 0 (0) | 4 (36.5) | 4 (50) | 1 (11.1) | 1 (14.3) |
| Pulmonary Function Measurements | | | | | | |
| Mean FVC % predicted at 3 months (SD) /Delta mean FVC % predicted (SD) 6 vs. 3 months | 96.2 (15.6) / 1.39 (7.1) | 103.5 (16.04) / 1.06 (5.37) | 96.3 (13.4) / 2.9 (7.7) | 82 (12.5) / 1.25 (11.2) | 91.6 (10) / 1.6 (6.1) | 101.42 (14.1) / -0.3 (5.92) |
| Mean DLCO % predicted at 3 months (SD)/Delta mean DLCO % predicted (SD) 6 vs. 3 months | 73.74 (18.0) / -3.14 (8.95) | 86.5 (9.5) / -2.3 (7.2) | 77.6 (14.5) / -5.1 (13.2) | 61.7(11.7) / -6.25 (4.96) | 65.5 (13.9) / -4.5 (7.1) | 71.4 (4.7) / 3.29 (5.92) |
| Mean VO2% predicted at 3 months (SD) /Delta mean VO2% predicted (SD) 6 vs. 3 months | 75.5 (18.9) / 0.57 (13.26) | 79.8 (17.4) / 4.5 (10.3) | 76.3 (37.6) / -0.36 (11.9) | 71.9 (23.5) / -2.34 (12.98) | 66.4 (16.1) / -0.33 (19.5) | 80.54 (17.8) / -2 (8.7) |

[a] COVID Severity Scale COVID-19 status during acute phase 1 = No medical intervention; 2 = ED visit; 3 = Hospitalized, not requiring non-invasive ventilation or use of high-flow oxygen devices4 = Hospitalized, requiring supplemental oxygen (nasal cannula or NRB mask (O2 flow rate in L/min); 5 = High flow heated oxygen (high flow nasal cannula (FIO2% reported) → HFNC; 6 = Hospitalization requiring intubation and ventilator

[b] Value obtained at 6-month visit

[c] 16 Patients had Asthma in their medical history prior to COVID-19 infection, and one patient with asthma was also diagnosed with COPD.

[d] denominator is 50 patients, as one case is missing SGRQ data

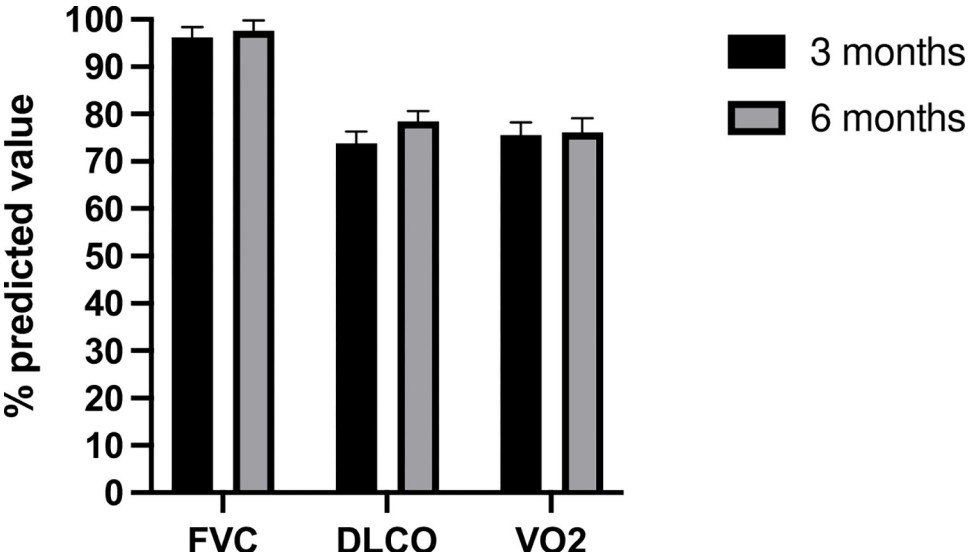

**Fig 2. Percent predicted of forced vital capacity (FVC), diffuse capacity of lung for carbon monoxide (DLCO), and oxygen consumption (VO2) at 3—and 6- months following COVID-19 infection.** Data is presented as mean and standard error of the mean.

persistent lung disease at 6 months (OR 9.75, 95% CI 1.74–54.79). Patients with elevated D-Dimer levels >250 ng/L, and those with hypertension and diabetes also had an increased odds of persistent lung disease at 6 months (Table 2). Pre-existing lung disease was not associated with persistent COVID lung disease at 6 months in the unadjusted analysis.

## Discussion

Our study reports a 6-month follow-up data on respiratory status of patients with COVID-19 infection at all severity levels in a single-centre Canadian population. In this cohort,

**Table 2. Clinical factors associated with persistent COVID-19 related respiratory abnormalities at 6 months post-infection^, using univariate logistic regression analyses, n = 51.**

| Clinical Predictors | Unadjusted Odds Ratio (95% Confidence Interval) |
|---|---|
| Age $>$ = 65 years | 1.71 (0.29–10.30) |
| Female Sex | 0.92 (0.19–4.35) |
| BMI $>$ = 30 kg/m$^2$ | 0.97 (0.17–5.53) |
| ESR $>$ = 20 | 1.71 (0.29–10.30) |
| CRP $>$ = 3 | 1.26 (0.22–7.33) |
| D-Dimer $>$ = 250 ng/L | 10.5 (1.18–93.70) |
| ICU Admission | 6.17 (1.21–31.55) |
| Pre-Existing Asthma/COPD | 0.69 (0.12–3.87) |
| Heart Disease | 6.00 (0.36–107.43) |
| Hypertension | 5.14 (1.03–25.60) |
| Diabetes | 9.75 (1.74–54.79) |
| Chronic Kidney Disease | 6.00 (0.36–107.43) |
| Cancer | 1.71 (0.29–10.30) |

^ There were 8 patients with persistent COVID-related lung disease, defined by clinician interpretation of chest radiograph findings, lung function data, CPET results, and clinical presentation at 6 months (Group 3 in Table 1).

approximately a third (19/51, 37%) (Groups 2 and 3) had physiologic and/or radiographic evidence of COVID lung disease at 3 months with almost half of this (8/19, 42%) or 15.7% of the whole study cohort continuing to have persistent disease at 6 months (Group 3). There was no improvement in lung function during the study period among patients with persistent lung disease at 6 months. Indeed, there was a slightly worsening in DLCO in this patient subgroup. Disease severity (ICU admission), elevated D-Dimer measurements, hypertension, and diabetes were associated with a significantly greater likelihood of having persistent lung disease at 6 months in unadjusted analyses.

A prospective multicentre trial reported a vast improvement of symptoms and CT abnormalities over time with sequential evaluations at 60 and 100 days after COVID-19 onset [9]. However, it is not known what the degree of improvement would be after 100 days or 3 months. Based on our study result, abnormal lung function at 3 months would resolve only in 58% of patients at 6 months. This indicates that lung function improvement occurs more in the earlier phase post-acute infection. In another study that only included hospitalized COVID-19 patients, 54% and 16% of patients had low DLCO (<80% predicted) and low FVC (<80% predicted) at month 6, respectively [10]. These numbers were higher than in our study cohort. However, of the 8 patients who had persistent lung disease at 6 months, all had severe COVID-19 needing hospitalization during the acute phase. Interestingly, the average % predicted DLCO and % predicted FVC (reported as median 76%, IQR 68–90, and 94%, IQR 85–104, respectively) were comparable to our cohort. This indicates data consistency and that a significant proportion of patients particularly those with severe COVID-19 infection will have persistent COVID-19 lung disease although the degree of impairment is mild.

We noted that a majority of patients with PFT abnormalities attributable to pre-existing lung disease or other factors reported worsened health-related quality of life by SGRQ at 6 months. This suggests that although these patients did not have new or worsening PFT and/or chest imaging following COVID-19 infection, they could still have worsening symptoms.

We acknowledge several major limitations in this study. Our study population was derived in a single centre, with non-consecutive enrolment. It is thus susceptible to referral bias. Subjects with mild disease and no long-term symptoms may choose not to enroll or to drop out after the 4 months assessment. Conversely, subjects who did not survive their COVID-19 infection were not studied.

Our sample size and number of outcomes was too small to permit a multivariate analysis. The univariate analyses for risk factors associated with persistent lung disease was performed post-hoc and therefore should be considered exploratory only.

The group assignments were based on a combination of physiologic testing, radiographic imaging and clinical assessment and may be imperfect. The groups were created to help categorize the different clinical trajectories in patients post-COVID infection. Furthermore, although all patients in groups 2–5 had radiographic imaging, this was limited to chest x-rays in some subjects and the timing of radiographic imaging was not specified in the original study protocol. This was due to the reduced availability of imaging resources at the time of our study.

It is possible that persisting COVID related lung disease could be contributory to the PFT abnormalities in groups 4 and 5, even in the absence of parenchymal findings on CXR.

It is important to acknowledge that all patients enrolled were infected with the original strain of COVID-19. Subsequent strains (Alpha and Delta) were associated with greater likelihood of severe disease and consequently may be associated with a greater likelihood of persistent lung disease. Furthermore, the study was performed at a time when treatment protocols (e.g., dexamethasone) had not been implemented and vaccines and antiviral therapies were not available. These factors may also influence clinical outcomes.

## Conclusion

Our study demonstrates that 16% of COVID-infected individuals had lung function and/or radiographic abnormalities at 6 months post infection. Furthermore, lung function for patients with persisting lung disease appeared to be stable between 3- and 6-months post infection, suggesting the possibility of longer-term respiratory deficits in some patients. Hospitalization as well as presence of comorbid diabetes may be risk factors for persistent COVID related physiologic abnormalities.

Our results highlight the need for longer term follow-up of patients with lung function abnormalities post-COVID to gain a better understanding of long-term physiological and functional outcomes.

## Supporting information

**S1 Checklist. Reporting checklist for cohort study.**
(DOCX)

**S1 Data.**
(XLSX)

## Acknowledgments

We would like to thank Dr. Sara Abdallah for her contribution on data collection and initial study administration, staff at the PFT and CPET labs for accommodating this research study, and staff at the Clinical Investigation Unit of the Ottawa Hospital Research Institute where the majority of the study took place.

## Author Contributions

**Conceptualization:** Juthaporn Cowan.

**Data curation:** Bashour Yazji.

**Formal analysis:** Bashour Yazji, Nha Voduc, Sunita Mulpuru, Juthaporn Cowan.

**Funding acquisition:** Juthaporn Cowan.

**Investigation:** Nha Voduc, Juthaporn Cowan.

**Methodology:** Nha Voduc, Juthaporn Cowan.

**Project administration:** Bashour Yazji, Juthaporn Cowan.

**Resources:** Juthaporn Cowan.

**Supervision:** Juthaporn Cowan.

**Validation:** Juthaporn Cowan.

**Writing – original draft:** Bashour Yazji, Juthaporn Cowan.

**Writing – review & editing:** Bashour Yazji, Nha Voduc, Sunita Mulpuru, Juthaporn Cowan.

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
