## [Decision Letter · Decision Letter 0]

4 Jul 2022

PONE-D-22-14376Pulmonary sequelae of SARS-CoV-2 infection and factors associated with persistent abnormal lung function at six months after infection: prospective cohort studyPLOS ONE

Dear Dr. Cowan,

Thank you for submitting your manuscript to PLOS ONE. After careful consideration, we feel that it has merit but does not fully meet PLOS ONE’s publication criteria as it currently stands. Therefore, we invite you to submit a revised version of the manuscript that addresses the points raised during the review process.

Based on the advice received, the Editor feels that your manuscript could be reconsidered for publication should you be prepared to incorporate major revisions.

When preparing your revised manuscript, you are asked to carefully consider the reviewer comments which are attached, and submit a list of responses to the comments.

Your list of responses should be uploaded as a file in addition to your revised manuscript.

We look forward to receiving your revised manuscript.

Kind regards,

Rano Mal Piryani, MBBS, MCPS, DTCD, MD, Fellowship in Med Education

Academic Editor

PLOS ONE

Journal Requirements:

Reviewers' comments:

Reviewer's Responses to Questions

**Comments to the Author**

1. Is the manuscript technically sound, and do the data support the conclusions?

Reviewer #1: Yes

Reviewer #2: Partly

Reviewer #3: Yes

2. Has the statistical analysis been performed appropriately and rigorously? 

Reviewer #1: Yes

Reviewer #2: No

Reviewer #3: Yes

3. Have the authors made all data underlying the findings in their manuscript fully available?

Reviewer #1: Yes

Reviewer #2: Yes

Reviewer #3: Yes

4. Is the manuscript presented in an intelligible fashion and written in standard English?

Reviewer #1: Yes

Reviewer #2: Yes

Reviewer #3: No

5. Review Comments to the Author

Reviewer #1: Thank you for allowing me to review this article. It is an interesting study ,no doubt although the sample size is small. I have a few questions for the authors which I hope they would clarify.

1.Which PfT machine was used for the study? Besides a restrictive abnormality ,was there an obstructive pattern on the PFT and if so was it reversible?

2.Were any of these patients on oral/inhaled steroids which could have affected outcome of the study?

3.Were any of these patients on a pulmonary rehabilitation programmme which again could have affected the results?

4.Was TLC done in these patients?

5.Was there a 2 D Echo and ABG for patients with severe impairment on PFT?

I thought the study had a few limitations.

1.The sample size was small

2.People with a pre existing lung disease could have been excluded from the study.

3.There was no base line PFT prior to contracting Covid in these patients.

4. Body plethesmography was not done in this study.

Regards.

Reviewer #2: Certain general pointers:

The study has very few participants and further smaller numbers in hospitalized and non hospitalized patients, making it difficult to draw any significant conclusions. Possibly the authors could have enrolled a larger cohort.

Radiology should have been available for the entire cohort at baseline and follow up to correlate with lung function abnormalities. The authors have not described extent of lung involvement on radiology at baseline or follow up.

The authors should have used the WHO COVID-19 severity scale which is a standardised scale for classifying the cohort rather than creating another scale. Also Line 85-87: To clear Grade 4 and 5 Line 92 : Could the authors describe NV

Patients with pre-existing lung diseases should have been ideally excluded from the cohort to see the exclusive effect of COVID-19 infection on radiology and spirometry.

Although the authors collected blood samples at 3/6 month visits, no mention of comparison between these tests in the different groups.

This study does not add any novelty to our already known knowledge - similar studies with larger cohorts have been published

McGroder CF, et al. Thorax 2021

Guler SA, et al. ERJ 2021

Evans RA, et al. Lancet Respiratory Medicine 2021

Sonnweber T, et al. ERJ 2021

Reviewer #3: Dear Author you have raised a concern on good topic ,however methodology is not written properly

please address the following issues

Expand all the abbreviations eg NV

Give flow chart of study population enrolled in the study

What kind of PFT have you used in study and describe the PFT abnormalities like obstruction restriction .and small airway disease .

what criterion have you chosen for pet abnormality .

Discussion part should be rewrite to understand properly

what are the radiological abnormality seen in lung diseases with PFT abnormal groups

6. PLOS authors have the option to publish the peer review history of their article (what does this mean?). If published, this will include your full peer review and any attached files.

Reviewer #1: No

Reviewer #2: **Yes: **Radhika Banka

Reviewer #3: No

---

## [Author Response · Author response to Decision Letter 0]

25 Sep 2022

We would like to express our gratitude for valuable comments. Below is our point-by-point response. 

Reviewer #1: Thank you for allowing me to review this article. It is an interesting study ,no doubt although the sample size is small. I have a few questions for the authors which I hope they would clarify.

1.Which PfT machine was used for the study? Besides a restrictive abnormality ,was there an obstructive pattern on the PFT and if so was it reversible?

Author response: We used Vmax systems made by SensorMedics. Pulmonary function testing was performed in accordance with American Thoracic Society standards and included spirometry, lung volume and diffusing capacity measurements. Yes, there was one participant who had obstructive lung pattern on PFT and was not reversible consistent with the known preexisting COPD. 

2.Were any of these patients on oral/inhaled steroids which could have affected outcome of the study?

Author response: Yes, 25.4% of enrolled participants were on inhaled corticosteroids for their pre-existing lung conditions. Our study was conducted early in the pandemic when systemic corticosteroids was not yet a standard of care for COVID-19 infection. We also did not observe any differences in FEV1, FVC, TLC and VO2 values among those who were on inhaled corticosteroids. 

3.Were any of these patients on a pulmonary rehabilitation programmme which again could have affected the results?

Author response: There were 5 participants who completed a pulmonary rehabilitation program as inpatients. The patients who underwent rehabilitation had a statistically significant increase in mean FVC and FEV1 values between 3 and 6 months as compared to the rest of the cohort (11.5% and 10.4% vs 0.5% and -0.6%, respectively). However, pulmonary function in patients who had rehabilitation was significantly lower than those who did not have rehabilitation as expected. 

4.Was TLC done in these patients?

Author response: Yes, a full pulmonary function test was done. One of the authors (Dr. Nha Voduc) took TLC data into consideration for PFT data interpretation. 

5.Was there a 2 D Echo and ABG for patients with severe impairment on PFT?

Author response: Yes, 2D echocardiography was also performed at month 3 visit as part of the study. Data was previously published (https://www.atsjournals.org/doi/10.1513/AnnalsATS.202012-1489RL). ABG was not done in our patient cohort as it is not a routine test for all abnormal PFT. ABG is typically done in certain cases such as patients with hypoxemia. None of our patients demonstrated significant desaturation during the cardiopulmonary exercise testing.

I thought the study had a few limitations.

1.The sample size was small

Author response: We agree and acknowledge this in our discussion. Thank you. 

2.People with a pre-existing lung disease could have been excluded from the study.

Author response: We did not exclude patients with pre-existing lung disease because we would like to observe if there was worsening existing lung disease following COVID-19 infection and to what extent. 

3.There was no base line PFT prior to contracting Covid in these patients.

Author response: Yes, it was unfortunate to not have baseline PFT. However, it was almost impossible to have baseline PFT data on healthy individuals before COVID-19 infection. 

4. Body plethesmography was not done in this study.

Author response: This was done. We clarified this in the revised manuscript. 

Reviewer #2: Certain general pointers:

The study has very few participants and further smaller numbers in hospitalized and non hospitalized patients, making it difficult to draw any significant conclusions. Possibly the authors could have enrolled a larger cohort.

Author response: We agree. We had limited resources during the pandemic to complete full PFT and CPET in a larger cohort. 

Radiology should have been available for the entire cohort at baseline and follow up to correlate with lung function abnormalities. The authors have not described extent of lung involvement on radiology at baseline or follow up.

Author response: All patients with abnormal lung function had corresponding radiology. Additional information regarding radiographic description was added in the revised manuscript on page 13. 

The authors should have used the WHO COVID-19 severity scale which is a standardised scale for classifying the cohort rather than creating another scale. Also Line 85-87: To clear Grade 4 and 5 Line 92 : Could the authors describe NV

Author response: The study was conceptualized very early in the pandemic. At the time, WHO COVID-19 severity scale was not yet available. 

We clarified definition of grade 4 and grade 5 on line 92. 

We clarified “NV” as one of our co-authors who is a pulmonology consultant. 

Patients with pre-existing lung diseases should have been ideally excluded from the cohort to see the exclusive effect of COVID-19 infection on radiology and spirometry.

Author response: We did not exclude patients with pre-existing lung disease because we would like to observe if there was worsening existing lung disease following COVID-19 infection and to what extent. 

Although the authors collected blood samples at 3/6 month visits, no mention of comparison between these tests in the different groups.

Author response: We analyzed biomarkers including ESR, D-dimer and reported the results in Table 2.

This study does not add any novelty to our already known knowledge - similar studies with larger cohorts have been published

McGroder CF, et al. Thorax 2021

Guler SA, et al. ERJ 2021

Evans RA, et al. Lancet Respiratory Medicine 2021

Sonnweber T, et al. ERJ 2021

Author response: Thank you for indicating these references. We cited some of these on our manuscript. 

These studies only included hospitalized patients except Guler SA, et al. ERJ 2021. Our study included both hospitalized and non-hospitalized patients. Although we showed similar results that approximately 80% of hospitalized patients developed persistent lung disease at 6 months, our study results added that persistent lung disease at 6 months did not occur in non-hospitalized patients in our study cohort. Thus, it is unlikely that non-hospitalized COVID-19 patients would have significant abnormal lung function at 6 months. Additionally, Guler et al. only reported data on symptoms and respiratory impairment at 4 months after infection while our study reported a longer-term data and provided clinical trajectory from 3 months to 6 months post infection. Therefore, despite similarities with other published articles, our study provides additional knowledge to the existing evidence. 

Reviewer #3: Dear Author you have raised a concern on good topic ,however methodology is not written properly

please address the following issues

Expand all the abbreviations eg NV

Author response: Apologies. We clarified that NV is an initial of one of the co-authors. 

Give flow chart of study population enrolled in the study

Author response: Thank you. Flow chart is added. 

What kind of PFT have you used in study and describe the PFT abnormalities like obstruction restriction .and small airway disease .

Author response: Pulmonary function testing was performed in accordance with American Thoracic Society standards and included spirometry, lung volume and diffusing capacity measurements.

what criterion have you chosen for pet abnormality .

Author response: We only reported peak VO2 data in this manuscript. Jones reference equations were used to determine predicted VO2 maximums for each subject. 

Discussion part should be rewrite to understand properly

Author response: We are uncertain about this comment as it is broad. We believe the discussion reflects the study data and revised as appropriate. 

what are the radiological abnormality seen in lung diseases with PFT abnormal groups

Author response: Thank you. We added information on radiological abnormality on page 13.

---

## [Decision Letter · Decision Letter 1]

1 Nov 2022

Pulmonary sequelae of SARS-CoV-2 infection and factors associated with persistent abnormal lung function at six months after infection: prospective cohort study

PONE-D-22-14376R1

Dear Dr. Cowan,

We’re pleased to inform you that your manuscript has been judged scientifically suitable for publication and will be formally accepted for publication once it meets all outstanding technical requirements.

Kind regards,

Rano Mal Piryani, MBBS, MCPS, DTCD, MD, Fellowship in Med Education

Academic Editor

PLOS ONE

Additional Editor Comments (optional):

Reviewers' comments:

Reviewer's Responses to Questions

**Comments to the Author**

1. If the authors have adequately addressed your comments raised in a previous round of review and you feel that this manuscript is now acceptable for publication, you may indicate that here to bypass the “Comments to the Author” section, enter your conflict of interest statement in the “Confidential to Editor” section, and submit your "Accept" recommendation.

Reviewer #1: All comments have been addressed

Reviewer #3: All comments have been addressed

2. Is the manuscript technically sound, and do the data support the conclusions?

Reviewer #1: Yes

Reviewer #3: Yes

3. Has the statistical analysis been performed appropriately and rigorously? 

Reviewer #1: Yes

Reviewer #3: Yes

4. Have the authors made all data underlying the findings in their manuscript fully available?

Reviewer #1: Yes

Reviewer #3: Yes

5. Is the manuscript presented in an intelligible fashion and written in standard English?

Reviewer #1: Yes

Reviewer #3: Yes

6. Review Comments to the Author

Reviewer #1: I have gone through the comments of the authors and found them to be satisfactory.Hence , the editorial board may proceed to do the needful.

Reviewer #3: (No Response)

7. PLOS authors have the option to publish the peer review history of their article (what does this mean?). If published, this will include your full peer review and any attached files.

Reviewer #1: No

Reviewer #3: No

---

## [Editor Report · Acceptance letter]

8 Nov 2022

PONE-D-22-14376R1 

Pulmonary sequelae of SARS-CoV-2 infection and factors associated with persistent abnormal lung function at six months after infection: prospective cohort study 

Dear Dr. Cowan:

I'm pleased to inform you that your manuscript has been deemed suitable for publication in PLOS ONE. Congratulations! Your manuscript is now with our production department. 

Kind regards, 

on behalf of

Dr. Rano Mal Piryani 

Academic Editor

PLOS ONE